# Changes in *Hox* Gene Chromatin Organization during Odontogenic Lineage Specification

**DOI:** 10.3390/genes14010198

**Published:** 2023-01-12

**Authors:** Gokul Gopinathan, Xinmin Zhang, Xianghong Luan, Thomas G. H. Diekwisch

**Affiliations:** 1Center for Craniofacial Research and Diagnosis, Texas A&M University, Dallas, TX 75246, USA; 2Bioinforx Inc., 510 Charmany Dr#275a, Madison, WI 53719, USA

**Keywords:** *Hox* genes, craniofacial, histone methylation, epigenetic, EZH2, odontogenic, neural crest

## Abstract

Craniofacial tissues comprise highly evolved organs characterized by a relative lack of expression in the HOX family transcription factors. In the present study, we sought to define the epigenetic events that limit *HOX* gene expression from undifferentiated neural crest cells to semi-differentiated odontogenic progenitors and to explore the effects of elevated levels of *HOX*. The ChIP-chip data demonstrated high levels of repressive H3K27me3 marks on the *HOX* gene promoters in ES and cranial neural crest cells when compared to the H3K4me3 marks, while the K4/K27 ratio was less repressive in the odontogenic progenitors, dental follicle, dental pulp, periodontal ligament fibroblasts, alveolar bone osteoblasts, and cementoblasts. The gene expression of multiple *HOX* genes, especially those from the *HOXA* and *HOXB* clusters, was significantly elevated and many times higher in alveolar bone cells than in the dental follicle cells. In addition, the *HOX* levels in the skeletal osteoblasts were many times higher in the trunk osteoblasts compared to the alveolar bone osteoblasts, and the repressive mark H3K27me3 promoter occupancy was substantially and significantly elevated in the alveolar bone osteoblasts when compared to the trunk osteoblasts. To explore the effect of elevated *HOX* levels in craniofacial neural crest cells, *HOX* expression was induced by transfecting cells with the Cdx4 transcription factor, resulting in a significant decrease in the mineralization markers, RUNX2, OSX, and OCN upon *HOX* elevation. Promoting *HOX* gene expression in developing teeth using the small molecule EZH2 inhibitor GSK126 resulted in an increased number of patterning events, supernumerary cusp formation, and increased Hoxa4 and Hoxb6 gene expression when compared to the controls. Together, these studies illustrate the profound effects of epigenetic regulatory events at all stages of the differentiation of craniofacial peripheral tissues from the neural crest, including lineage specification, tissue differentiation, and patterning.

## 1. Introduction

The vertebrate head is a highly innovative structure in the evolution of deuterostomes that gave the early vertebrates significant advantages in terms of predation, mastication, and respiration, facilitating successful competition for novel food sources [1,2]. From a developmental perspective, Gans and Northcutt’s “new head” coincides with the emergence of a cranial population of neural crest cells, the cranial neural crest cells (CNCCs), and the limitation of *HOX* gene expression to the caudal regions of the first pharyngeal arch and along the vertebrate body axis. In contrast, CNCCs migrating into the “new head” do not express *HOX* genes [3]. The essential role of the loss of *HOX* gene expression for craniofacial pattern specification has been demonstrated in an elegant set of experiments in which a CNCC-deficient phenotype was rescued with a graft from a *HOX*-negative tissue, while a *HOX*-positive graft was unable to rescue the phenotype [4,5,6]. These studies establish the boundary between *HOX* code-defined patterning along the vertebrate body axis from the caudal extension of the vertebrate column toward the second pharyngeal arch, while the patterning rostral of the first pharyngeal arch and into the vertebrate head is largely defined by non-*HOX* transcription factors, such as Msx, Dlx, and others. The specific absence of *HOX* gene expression in CNCCs within the first pharyngeal arch and anterior domains is crucial for the formation of the chondrogenic and skeletal elements of the facial region [4,7,8]. A correlation between the regulation of *HOX* gene expression in CNCCs and the bone and cartilage formation in the craniofacial skeleton was also demonstrated in mice subjected to conditional inactivation of the Polycomb repressive complex subunit *Ezh2* [9]. A direct link between *HOX* gene repression in CNCCs and the bone and cartilage formation in the craniofacial skeleton was also demonstrated in mice subjected to conditional inactivation of the Polycomb repressive complex subunit, *Ezh2* [9]. The strong upregulation of *HOX* genes in these *Ezh2* knockout mice resulted in the suppression of the osteochondrogenesis program and a dramatic reduction in the expression of the osteogenesis markers, RUNX2, OSTERIX (OSX), and ALP [9]. More recently, it was demonstrated that the ectopic expression of *Hoxa2* in mice led to a complete loss of craniofacial skeletal structures, whereas the overexpression of *Hoxa5* led to a less severe craniofacial skeletal defect [10].

*HOX* genes are a large family of transcription factors that comprise a 120-nucleotide homeobox, which is part of a 61 amino acid helix-loop-helix motif, the homeodomain DNA-binding motif. The homeobox was originally discovered as a shared sequence element of about 180 bp in the homeotic genes in Drosophila melanogaster, which gave rise to its name [11,12]. Homeodomain proteins act as regulatory genes in processes such as pattern formation and the evolution of the patterning mechanism [13]. Genome duplications during vertebrate evolution have given rise to four *HOX* clusters: *HOX* A, B, C, and D [14]. The effect of *HOX* genes on the patterning along the mouse anterior–posterior axis has been explained as being either the result of varying combinations of *HOX* genes at any given axial level [15] or through phenotypic suppression, by which posterior *HOX* genes are functionally dominant over the anterior genes [16]. The *HOX* gene function is required for the morphogenesis of a broad variety of tissues, including neural tissues, neural crest, endoderm, and mesodermal derivatives [17,18,19,20]. Previous studies reference that “the mandible and maxilla do not express *HOX* genes, the anterior border of which sits between the first and second pharyngeal arch” [21]. Instead, patterning in the mandible is achieved by unique craniofacial transcription factors, such as LHX6,7; GSC; MSX1,2; DLX1,2; BARX1; and PITX1 [22,23,24,25]. 

In pluripotent stem cells, the *HOX* genes are silenced and display bivalent chromatin marks [26]. It is thought that at this stage epigenetic marking prepares the *HOX* genes for future activation [26]. As differentiation proceeds, the bivalent marks are resolved, and the relevant developmental genes retain permissive marks and lose repressive marks [27,28]. Active *HOX* genes are characterized by H3K4 acetylation marks for euchromatin accessibility [27,28]. Once cells are fully differentiated and developmental programs are completed, *HOX* genes are generally occupied by repressive H3K27me3 marks [29,30,31]. The epigenetic control of *HOX* gene expression is achieved through histone methyltransferases of the Polycomb (PcG) and Trithorax (TrxG) group [28]. PcG proteins are transcriptional repressors that silence target gene expression through the PRC 1 and PRC 2 protein complexes [32]. The PRC 2 complex comprises a number of individual protein components, including EZH2 and SUZ12, both of which can be used as reagents for individual function studies [28]. Both the Polycomb and the Trithorax group of proteins have also been implicated in the deregulation of *HOX* genes in human cancers [33,34], illustrating both their importance and their potency for dedifferentiation. 

The present study was prompted by the outcomes of a series of ChIP-seq and ChIP-chip studies identifying a remarkable repression gradient in *HOX* genes ranging from embryonic stem cells to differentiated periodontal progenitors and other lineages. These data led to the hypothesis that the gene expression during craniofacial lineage commitment was to a large extent the result of fine-tuned levels of lineage-specific epigenetic repression. To ask whether *HOX* genes were poised for expression during odontogenic lineage specification, we compared the relative abundance of the active and repressive histone modifications between CNCCs and various odontogenic progenitors. Specifically, we compared epigenetic modifications on the *HOX* gene promoters of dental follicle and alveolar bone cells and linked histone modifications to expression levels. To examine the effect of *HOX* gene overexpression on neural crest lineage differentiation, *HOX* gene overexpression via Cdx4 was induced, and its effect on mineralization marker gene expression was determined. To assess whether *HOX* genes are differentially repressed in the craniofacial and trunk neural crest, repressive and active histone modifications in the alveolar bone and trunk osteoblasts were compared. Furthermore, developing tooth organs were treated with epigenetic small molecule inhibitor reagents against the H3K27me3 catalyst and PRC 2 complex member EZH2, alleviating the epigenetic silencing of the *HOX* function and activating *HOX*. Together, these studies provide a comprehensive survey of the effects of the epigenetic repression of *HOX* genes during mammalian development and a preliminary assessment of the consequences of aberrant *HOX* gene activation during mandibular development.

## 2. Materials and Methods

### 2.1. Isolation and Culture of Human Odontogenic Progenitors

Isolation of primary progenitor cells from odontogenic tissues was performed as described previously [35]. Briefly, dental follicle (DF), dental pulp (DP), periodontal ligament (PDL), and cementum (CEM) tissues were dissected from developing tooth organs obtained from healthy human teeth (from patients ranging from 12 to 15 years) extracted for orthodontic reasons. Alveolar bone (AB) cells were extracted from bone exostoses collected from healthy tooth extractions. Mesenchymal progenitors were isolated from dental tissues after digestion with collagenase/dispase, and the digested tissues were seeded onto culture dishes in complete media (1X DMEM containing 10% fetal bovine serum, 100 U/mL penicillin, 100 µg/mL streptomycin) for cellular outgrowth. The human osteoblasts (HOB) used in the study were from femoral tissue and were obtained from PromoCell (C-12720) and cultured in the manufacturer’s recommended media (C-27001). Confluent cultures were subcultured and used for experiments at an early passage (P2-P4). Collection of human tissues was performed in accordance with human subject protocol approved by Texas A&M University’s Institutional Review Board.

### 2.2. Mineralization Induction of DF, AB, and CNCC

The DF and AB progenitors were seeded at a density of 25,000 cells/cm^2^ and cultured for 24 to 48 h until confluency was attained. Mineralization was induced with osteogenic induction medium (complete media supplemented with 50 µg/mL ascorbic acid 2-phosphate, 10mM β-glycerophosphate, and 10 nM dexamethasone). The induction medium was changed every alternate day. The cells were collected for various experiments after 5 or 10 days of induction. The cells prior to induction were used as the reference group (control). O9-1 mouse cranial neural crest cells (CNCC) (SCC049, Millipore Sigma, St. Louis, MO, USA) were cultured in complete ES Cell Medium (ES-101-B, Millipore Sigma, St. Louis, MO, USA) supplemented with 25 ng/mL FGF-2 (GF003, Millipore Sigma) on MatrigelTM (CB-40234, Fisher Scientific) coated cell culture dishes. CNCC differentiation was induced in osteogenic differentiation media containing α-MEM, 10% FBS, 100 U/mL penicillin, 100 µg/mL streptomycin, 10 mM β-glycerophosphate, 50 µg/mL ascorbic acid 2-phosphate, 10 nM dexamethasone, and 100 ng/mL BMP2 (R&D Systems) and harvested for the experiments described above.

### 2.3. Chip-on-Chip Analysis

The ChIP-chip assay and initial bioinformatic analysis were performed essentially as described earlier [35]. Early passage DF, DP, PDL, AB, and CEM progenitors were used for the assay. Nuclei from formaldehyde-treated cells were lysed in cold lysis buffer and chromatin sonicated to a size of 300bp-1kb in a cup horn sonicator (Q Sonica, Newtown CT). Equal amounts of sheared chromatin were incubated overnight with 100 µL of DynaI beads (Invitrogen, Carlsbad, CA, USA) pre-bound to 10 µg of antibodies against H3K4me3 and H3K27me3 histone modifications (Abcam, Cambridge, MA, USA). An input fraction corresponding to 10% of the starting chromatin was kept aside for background normalization. The immunoprecipitated chromatin was washed five times with RIPA buffer and once with 1XTE, and the bound protein-DNA complexes were eluted by incubating with elution buffer at 65 °C, followed by crosslink reversal overnight at 65 °C. The DNA was purified by Proteinase K digestion, phenol-chloroform extraction and ethanol precipitation. The resulting pellet was resuspended in 50 µL of 10 mM Tris pH8.0. The ChIP experiments were performed as triplicates.

### 2.4. Sample Preparation for ChIP-Chip, Whole Genome Amplification, Dual-Color Labeling, and Array Scanning

The ChIP DNA quality and concentration was determined on a NanoDrop spectrophotometer. All the samples were whole genome amplified. Dual-color labeling reactions were performed using the NimbleGen Dual-color labeling Protocol (ver 6.2). One microgram of the immunoprecipitated and the input DNA sample was labeled with Cy5 and Cy3, respectively. The labeled DNA was then purified, and the labeling efficiency was determined by the NanoDrop spectrophotometer. Fifteen micrograms of labeled IP DNA was pooled for hybridization with 15 µg of labeled input DNA, placed on the 3X720K Roche Human ChIP-chip Promoter microarray and hybridized at 42 °C for ~20 h. The arrays represent 22,542 promoters based on a human genome 18 build (HG18) with a tiling of 3200 bp upstream and 800 bp downstream to each transcription start site (TSS). Finally, the arrays were washed and scanned at a 2 µm resolution on a NimbleGen MS 200 Microarray Scanner. The data were processed through DEVA software using NimbleScan Software (ver 2.6) and the Design files from the genome annotations for HG18 Refseq promoters. A quality experimental metrics report was produced for each sample, which met the guidelines set by Roche NimbleGen.

### 2.5. Raw Data Processing and Gene Annotation

Datasets corresponding to the histone enrichment values in both the immunoprecipitated (IP) and the input samples were processed to obtain Ratio.GFF files (log_2_ IP/input). All the samples passed the QC metrics provided by Roche NimbleGen (Roche). The Ratio.GFF files from the triplicates for each experiment were merged to obtain average values for histone modification enrichment. These were then converted to wiggle files for visualization in the IGV genome browser. Initial peak calling was performed using NimbleScan software (Roche), and initial calls were further filtered to obtain significant peaks (log_2_ ratio > 1 and the false discovery rate (FDR) < 5%). The filtered peaks were subsequently mapped to overlapping features 5000 bp upstream and 1000 bp downstream of the nearest transcription start site (TSS), and a peak report was generated. The promoter information and coordinates were used for downstream analysis. 

### 2.6. Data Processing for Visualizing ChIP-Chip and ChIP-Seq Enrichment

The average values for the histone modification enrichment from the triplicates for each ChIP-chip experiment (log2 ChIP/input) from the DF, DP, PDL, AB, and CEM progenitors were merged and then converted to wiggle files for visualization in the IGV (Integrative Genomics Viewer) genome browser [36]. Normalized reads from the ChIP-seq data for bone marrow mesenchymal stem cells (MSC-GSE89179) and neural crest cells (CNCC-GSE28874) were sourced from the NCBI gene expression omnibus database and aligned to the hg19 genome version for IGV plots. The histone enrichment data for the embryonic stem cells (ES), human skeletal muscle myoblasts (HSMM), and normal human epidermal keratinocytes (NHEK) were obtained from the IGV server. Peak calling was performed using the MACS2 software and annotated using the PeakAnalyzer using the default set of parameters. The track height was set to auto scale in the IGV to accommodate the enrichment data from all the peak histograms. For the box plot visualization of the histone enrichment data across all the samples for each *HOX* cluster, the promoter signal values for each *HOX* gene were plotted relative to the highest enrichment value observed (100th percentile). The whiskers in each box plot represent the lowest and highest promoter signal value among the individual *HOX* genes, while the average is represented as a vertical line within the box. For the identification of promoters having a bivalent signature, the H3K4me3 and H3K27me3 peaks were examined for overlaps with at least one nucleotide base in common. These overlap sites were identified as potential bivalent domains and further mapped to individual promoters. For bivalent domain visualization at the *HOX* promoters, the log2 value of H3K4me3/H3K27me3 was computed for each promoter across all the samples. The bar plots with negative values are H3K27me3-rich; the positive values are H3K4me3-rich, while the values trending near zero on the scale are treated as “bivalent”. 

### 2.7. Chromatin Immunoprecipitation Analysis 

ChIP assays were performed using the Zymo-Spin ChIP kit (Zymo research, Irvine, CA, USA) as per the manufacturer’s instructions with a control and 5-day induced and 10-day induced DF, AB, and CNCC progenitor cells as experimental groups. H3K4me3 and H3K27me3 immunoprecipitations were carried out with an equal amount of chromatin, and bound DNA was eluted in a total volume of 16 µL. Quantitative PCR was performed on a Bio-Rad CFX96 Real-Time machine with 2 µL of ChIP DNA for each histone modification and sample. Enrichment values from the total input were used as internal reference for data normalization and the beads alone served as a negative control. Each ChIP PCR was performed on at least 4–6 individual experimental samples. Primer pairs (Appendix A) specifically designed against the promoter regions of *HOX* genes were used for q-PCR to quantify enrichment.

### 2.8. Cdx4 Plasmid Constructs and Transfections

Plasmids (pcDNA3.1+/C-(K)-DYK) encoding full-length mouse Cdx1, Cdx2, and Cdx4 were obtained from Genscript. O9-1 mouse cranial neural crest cells (CNCC, SCC049, Millipore Sigma) were transfected with Cdx constructs using EndoFectin Max transfection reagent (GeneCopoeia, Rockville, MD, USA). The transfected cells were split after 48 h and selected for stable plasmid integration using G418 antibiotic (Gibco, ThermoFisher Scientific, Waltham, MA, USA) for 8–10 days. Resistant cell colonies were subcultured and overexpression ascertained by q-PCR analysis before use in the experiments. 

### 2.9. RNA Extraction and RT PCR 

Total RNA was isolated using the RNeasy Plus Mini Kit (Qiagen) as per the manufacturer’s instructions and quantified on a Nanodrop. Two micrograms of total extracted RNA were applied toward cDNA generation with the RNA to cDNA EcoDry Premix kit (639549, TaKaRa). The primer sequences used for determining the expression of all *HOX* genes in the mouse [37] and human cells [38] were described previously. The primers for q-PCR were designed based on the NCBI/GenBank sequence database. Real-time PCR was performed using SYBR green Master Mix (Applied Biosystems, Waltham, MA, USA) and the Bio-Rad CFX96 Real-Time PCR system (Bio-Rad) and normalized with GAPDH levels for each sample. The analyses were performed in triplicate from more than 5 independent experiments to confirm the reproducibility of the results. Relative expression levels were calculated using the 2 ^−ΔΔCt^ method [39], and the values were graphed as the mean expression level ± standard deviation (SD). In some cases, the PCR products were electrophoresed on 2% agarose gels, stained with ethidium bromide, and visualized under UV light. All the primers used in the study are listed in Appendix A.

### 2.10. Immunohistochemistry

The tooth organs were fixed with 10% formalin, embedded in paraffin, and cut to a thickness of 6 µm. For immunohistochemistry, the sections were deparaffinized and rehydrated to water followed by heat-induced epitope retrieval using the pressure cooker method (TintoRetriever, Bio SB, Santa Barbara, CA, USA). Antibody staining was performed using the Vectastain Elite ABC Kit following the manufacturer’s instructions (PK-6101, Vector Laboratories, Burlingame, CA, USA). The sections were incubated with anti-HOXA4 antibody (PA579389, ThermoFisher Scientific) and anti-HOXB6 antibody (PA5116164, ThermoFisher Scientific) for 1 h at room temperature. Antibody staining was revealed using an HRP detection reagent (ImmPACT AMEC Red, Vector Laboratories). Stained sections were subsequently stained with Hematoxylin QS (Vector) and mounted using an aqueous media (Hydromount, Electron Microscopy Services, Hatfield, PA, USA). Masson’s Trichrome staining was performed using a Trichrome kit (HT15-1KT, MilliporeSigma, Burlington, MA, USA) as per the instructions in the kit. 

### 2.11. Mouse Tooth Organ Culture and EZH2 Inhibitor Treatment

Tooth organs from e16 embryos (obtained from C57BL/6 timed-pregnant females, Charles River) were dissected and late cap stage M1 molars were isolated. The organs were grown for 10 days in a Trowell Organ culture system using 1X BGJb medium (Fitton-Jackson, Gibco), 20% FBS, 100 µg/mL ascorbic acid, 100 U/mL penicillin, and 100 µg/mL streptomycin (Evans et al. 1998). The molar explants were oriented on a Millipore filter disc to identify left and right quadrant tooth organs. EZH2 inhibition was carried out by adding the selective inhibitor GSK126 (5 µM final concentration). Enamel thickness measurements were carried out using Image J software (NIH, v1.52a). A minimum of 50 measurements were performed for each experimental condition.

### 2.12. Statistical Analysis

The quantitative data are summarized as mean ± SD (Standard Deviation) and compared using t test (Graph Pad Prism v 8.0, Graph Pad, San Diego, CA, USA). The difference between groups was considered statistically significant (*) at *p* < 0.05. The sample size is mentioned in the corresponding figure legends.

## 3. Results

### 3.1. Repressive Histone Modifications Marked by H3K27me3 at HOX Promoters Distinguish Craniofacial Mesenchymal Progenitors from Trunk Lineages 

The dental tissues residing within the periodontium are derived from cranial neural crest cells (CNCCs) reported to lack *HOX* gene expression [40,41,42,43]. To identify the underlying epigenetic marks regulating *HOX* gene expression within the periodontium, we performed a comprehensive analysis of the histone methylation profiles for all four human *HOX* clusters (*HOX* A, B, C, and D) in the odontogenic lineage dental follicle cells (DF), dental pulp progenitors (DP), periodontal ligament progenitors (PDL), alveolar bone osteoblasts (AB) and cementoblasts (CEM) using chromatin immunoprecipitation followed by promoter Chip hybridization (ChIP-chip). Enrichment profiles for the active histone methylation mark, H3K4me3, and the repressive histone methylation mark, H3K27me3, were compared based on their key roles in regulating gene expression during lineage commitment and the differentiation of embryonic and mesenchymal stem cells [44,45,46]. Immunoprecipitated DNA for H3K4me3 and H3K27me3 antibodies from early passage human DF, DP, PDL, AB, and CEM primary cells were hybridized onto a tiling array probing a promoter region between −3.2 kb and +0.8 kb relative to the transcription start site (TSS) of 22,542 human promoters. Our analysis indicated a high level of correlation and reproducibility between replicate datasets for both histone modifications in all five cell types analyzed. 

Using the IGV (Integrative Genomics Viewer) browser [36], the average log_2_ signal ratio of ChIP/input from the replicates for each histone modification on the individual *HOX* gene promoters was compared between odontogenic progenitors (Figure 1A–D). 

As a reference for *HOX* gene signatures representative of the non-CNCC-derived cell lineages, histone methylation data from curated online databases (NCBI) were obtained for two multipotent cell types, mesenchymal stem cells (MSC) and human skeletal muscle myoblasts (HSMM), and embryonic stem (ES) cells as a pluripotent lineage reference. We also included histone H3K4 and H3K27 methylation data for the CNCCs obtained from the online database (NCBI) in our comparisons. Our analysis indicated that ES cells and CNCC cells featured the highest levels of repressive H3K27me3 histone modifications across the entire of the *HOX* A, B, C, and D clusters, with considerably lower enrichment for the H3K4me3 modification marks (Figure 1A–D). Among the five dental lineages, the DF, DP, and PDL cells exhibited higher levels of H3K27me3 enrichment on several *HOX* promoters within the *HOX* A and *HOX* B clusters compared to the AB and CEM cells. Specifically, the H3K27me3 levels were reduced for *HOXA2*, *A3*, and *A4* within the *HOX* A cluster and for HOXB1, B2, B4, and B7 within the *HOX* B cluster in the AB and CEM progenitors compared to the DF, DP, and PDL progenitors (Figure 1A,B). On the other hand, the H3K27me3 levels on the *HOX* C and *HOX* D promoters did not differ substantially between the odontogenic progenitors studied (Figure 1C,D). Our analysis also demonstrated an overall high level of the H3K4me3 promoter signal on the *HOX* promoters from all four clusters among the odontogenic progenitor lineages (Figure 1A–D). Interestingly, the *HOX* gene promoters in the *HOX* A and *HOX* B cluster were almost exclusively enriched for H3K4me3 in both the MSCs and HSMMs while the *HOX* C and *HOX* D clusters featured a mixed H3K4me3/H3K27me3 enrichment profile across the entire cluster (Figure 1A–D). We also verified histone methylation at several other gene promoters with established enrichment patterns for H3K4me3 and H3K27me3, including the pluripotency-related factors, *Nanog* and *Sox2*, and the universally expressing gene, *Gapdh* (Appendix A). 

To negate the effect of autoscaling in our IGV visualizations, a comparative analysis of histone enrichment signals at the cluster level for the *HOX* A, B, C, and D clusters was conducted to reveal the true scale of the relative enrichments for H3K4me3 and H3K27me3 marks across all cell types (Figure 1E). The box plots demonstrated that within a *HOX* cluster matched to an individual cell type, the repressive H3K27me3 marks exhibited the maximum amount of enrichment, approaching the 100th percentile with minimal variability for ES, CNCCs, and all the odontogenic lineages tested (Figure 1E). In contrast, the H3K27me3 enrichment signals were highly variable and remarkably lower for all the *HOX* clusters in the MSCs and HSMMs, except for the *HOX* D cluster in the HSMM cells, which exhibited maximum enrichment levels. As observed with the IGV plots, the ES and CNCC cells revealed the lowest levels of H3K4me3 enrichment when compared to the odontogenic progenitors, while the promoter enrichment signals for MSC and HSMM were highly variable (Figure 1E). 

### 3.2. HOX Gene Promoters in Odontogenic Progenitors Were Characterized by a Bivalent Histone Methylation Signature 

*HOX* genes in ES cells are regulated by bivalent chromatin characterized by the presence of both H3K27me3 and H3K4me3 histone modifications, with H3K27me3 as the dominant modification [45]. 

Regardless of their high level of transcriptional repression, such bivalent domains remain in a poised state since gene expression may rapidly commence upon removal of the repressive histone marks [47]. The presence of both H3K27me3 and H3K4me3 histone modifications at the *HOX* promoters in our ChIP-chip analysis were indicative of a bivalent chromatin signature in odontogenic progenitors (Figure 1A–E). To verify the presence of a bivalent signature on the odontogenic *HOX* gene promoters, we compared the log2 ratio of the H3K4me3/H3K27me3 enrichment values for each promoter across all the cell types. 

Our analysis revealed a bivalent chromatin signature with enrichment for both histone marks at the *HOX* promoters across all four *HOX* clusters in the DF, DP, PDL, AB, and CEM cells, and all the promoters exhibited slightly higher H3K27me3 than H3K4me3 enrichment values (Figure 2A). As expected, the ES cell *HOX* promoters also demonstrated a bivalent signature, although with the relatively higher negative enrichment ratios indicative of the generally higher H3K27me3 enrichment observed across the *HOX* clusters in the ES cells (Figure 2A). In contrast, the H3K4me3/H3K27me3 ratios in the CNCCs were notably lower and trended toward a higher negative value compared to both the ES cells and the odontogenic progenitors (Figure 2A). In support of the results of our IGV plot analysis and box plot analyses (Figure 1A,B), the H3K27me3 enrichment was lost for the multiple *HOX* genes in the MSCs, with positive enrichment ratios for *HOX* promoters across all four clusters indicating higher H3K4me3 enrichment and a lack of H3K27me3 repressive modification. In addition, the analysis of *HOX* promoter bivalency in a highly committed cell lineage such as the normal human epidermal keratinocytes (NHEKs) revealed that numerous *HOX* promoters had lost their bivalent signatures, yielding either higher positive or negative enrichment ratios when compared to the odontogenic progenitors (Figure 2A). 

To further quantify the bivalent domains among the *HOX* promoters in odontogenic progenitors, we probed our ChIP-chip data to identify the total number of bivalent promoters in each cell type. For this analysis, H3K4me3 and H3K27me3 peaks with at least one common base pair in the overlap region were considered as potentially bivalent sites, which were then mapped to promoters. Our analysis identified a high number of bivalent promoters among the odontogenic progenitors, with the highest number of bivalent promoters in the AB cells (661 bivalent promoters) and the lowest number in the DP cells (270 bivalent promoters) (Figure 2B). This analysis also confirmed the presence of several *HOX* promoters among the bivalent promoters identified in all the odontogenic progenitors (*HOX* gene count: DF-5, DP-7, PDL-11, AB-8, and CEM-19) (Figure 2B). Considering the high prevalence of H3K27me3 marks on the *HOX* promoters, we next analyzed the transcript levels of EZH2, the enzymatic catalytic subunit of the Polycomb Repressive Complex 2 (PRC2) which mediates the trimethylation of H3K27. The EZH2 transcript levels were the highest in CNCCs, followed by the DP cells. Other odontogenic progenitors, including the DF, PDL, AB, and CEM cells, exhibited significantly lesser EZH2 transcript levels (Figure 2C). To validate our ChIP-chip findings, we conducted a ChIP assay for H3K4me3 and H3K27me3 enrichment at the HOXA10 promoter in the DF, PDL, AB, and CEM odontogenic progenitors. The *HOXA10* promoter was chosen as a representative *HOX* promoter based on our analysis, indicating a higher negative value for the H3K4me3/H3K27me3 ratio in the DF cells and a gradual increase for the H3K4me3/H3K27me3 ratio in the other odontogenic lineages, PDL, AB, and CEM. This assay demonstrated that both the H3K4me3 and the H3K27me3 histone marks were present on the *HOXA10* promoter (Figure 2D). While H3K4me3 enrichment on the *HOXA10* promoter was similar in all four cell types, the DF cells exhibited the highest level of enrichment for the H3K27me3 marks, and the AB cells exhibited significantly less enrichment for this repressive mark on the *HOXA10* promoter. 

### 3.3. Higher Levels of HOX Gene Expression and Reduced Levels of H3K27me3 Histone Marks in AB Cells Versus DF Cells

To gain further insights into the significance of *HOX* gene epigenetic silencing and its influence on gene expression, we compared the *HOX* gene expression levels across all four clusters (*HOXA*, *HOXB*, *HOXC*, and *HOXD*) in human DF and AB cells. These two odontogenic progenitors were chosen because both are CNCC derivatives, with DF progenitors being the least differentiated and AB cells being fully committed bone progenitors [48,49]. Real-time PCR analysis revealed that *HOX* genes belonging to the *HOXA* and *HOXB* cluster were significantly upregulated in the AB cells compared to the DF cells (Figure 3A), in line with our ChIP-chip observations, demonstrating higher H3K4me3 and lower H3K27me3 enrichment levels in the AB cells compared to the DF cells. The genes that were significantly elevated in AB cells included HOXA2, HOXA4, HOXA6, HOXA9, and HOXA10 from the *HOXA* cluster and HOXB2, HOXB3, and HOXB5 from the *HOXB* cluster, among others (Figure 3A). The *HOX* gene expression among the *HOXC* and *HOXD* clusters did not significantly differ between the DF and the AB cells, with the exception of HOXD10 and HOXD13, which were surprisingly expressed at lower levels in the AB cells compared to the DF cells.

To compare and correlate the relative levels of H3K4me3 and H3K27me3 histone marks between the DF and the AB cells with gene expression, H3K4me3 and H3K27me3 ChIP assays on individual *HOX* gene promoters were performed. These ChIP comparisons clearly demonstrated higher levels of H3K4me3 enrichment in tandem with lower H3K27me3 enrichment at the promoters of *HOXA4* and *HOXB2* in the AB cells compared to the DF cells (Figure 3B). Notably, both HOXA4 and HOXB2 featured among the most highly expressed *HOX* genes in the AB cells, thus providing a clear correlation between histone modification and gene expression at the *HOX* loci. There was a similar enrichment pattern with higher H3K4me3 and lower H3K27me3 marks at the *HOXC11* and *HOXD9* promoters in the AB cells when compared to the DF cells. Although these differences were not to the extent of what was observed for *HOXA4* and *HOXB2*, they were statistically significant, reaffirming our ChIP-chip analysis (Figure 3B). Notably, the ChIP enrichment data for the *HOXA4* promoter were not in accordance with our bioinformatic analysis following ChIP-chip. We believe this discrepancy in histone enrichment is a consequence of the differential placement of the ChIP PCR primer within the promoter region as opposed to the hybridization probes on the chip array and as such are not representative of the entire region. 

To identify the *HOX* genes that are upregulated during mineralization, *HOX* gene expression changes during in vitro mineralized lineage differentiation in both the DF and the AB cells were compared. For this study, the DF and AB cells were subjected to mineralization induction for 5 and 10 days, and transcript levels of *HOX* genes were determined using real-time PCR analysis. To our surprise, there were no changes in *HOX* gene expression among the DF cells over the 5-day or 10-day period of mineralization induction (Appendix A). In contrast, the AB cells revealed a significant upregulation of several *HOX* genes over the 10-day period of in vitro differentiation, featuring higher transcript levels for several *HOXA* cluster genes including, HOXA2, HOXA7, HOXA9, HOXA10, and HOXA11, and also HOXB9 from the *HOXB* cluster (Figure 3C). Interestingly, the *HOX* genes that were upregulated upon mineralization induction in the AB cells predominantly belonged to the *HOXA* cluster, with no gene expression differences observed among the *HOX* genes in the *HOXC* and *HOXD* clusters. The corresponding ChIP analysis to investigate changes in the H3K4me3 and H3K27me3 enrichment levels on the *HOX* promoters during in vitro mineralization induction in the DF and AB cells demonstrated a gradual and significant decrease in H3K4me3 levels after 5 days and 10 days, with a concomitant upregulation of H3K27me3 levels after 10 days of differentiation at the *HOXA2* and *HOXB9* promoters in the DF cells (Figure 3D,E). This decrease in H3K4me3 in our ChIP analysis might explain the lack of *HOX* gene upregulation in DF cells upon mineralization induction. There was also a significant reduction in H3K27me3 enrichment over the 10-day induction period for both the HOXA2 and the HOXB9 promoters in the AB cells, with no significant change in the overall levels of H3K4me3 after 10 days (Figure 3D,E). Together, these data suggest that AB cell differentiation coincides with reduced H3K27me3 repression and a resulting relative increase in *HOX* gene expression. 

### 3.4. Cdx4 Overexpression Induces HOX Gene Upregulation and Decreased Mineralization Marker Gene Expression in Mouse Cranial Neural Crest Cells

In comparison to trunk neural crest cells, craniofacial neural crest cells lack *HOX* gene expression [50]. To identify the effects of aberrant *HOX* gene activation in CNCC differentiation, we overexpressed the ParaHox transcription factors CDX1, CDX2, and CDX4 in CNCCs. The CDX family of transcription factors control *HOX* gene transcription directly through numerous CDX binding sites embedded within the *HOX* clusters [51]. For this study, mouse CNCCs were transfected with plasmids encoding full-length Cdx1,2, and 4 cDNAs and selected for cells stably expressing each of the Cdx constructs separately. Gene expression analysis in early passage CDX-expressing CNCCs indicated that CDX1,2, and 4 overexpression did not result in any significant changes in the expression levels of the CNCC markers, including CD44, Sox9, ScaI, and Nestin (Appendix A). 

However, the overexpression of both CDX1 and CDX4 in CNCCs resulted in *HOX* gene activation, while CDX2 overexpression did not result in *HOX* gene upregulation (Appendix A. We chose Cdx4-expressing CNCCs (CNCC/Cdx4) for our subsequent assays since the stable expression of Cdx4 in CNCC resulted in a higher and more uniform upregulation of *HOX* genes across all four *HOX* clusters compared to the Cdx1-expressing or the control CNCCs (CNCC/control). Specifically, real-time expression analysis revealed that Cdx4 overexpression significantly upregulated multiple *HOX* genes, including HOXA4, HOXA9, HOXA10, HOXB3, HOXB6, HOXC4, HOXC6, HOXD1, and HOXD13 in CNCCs (Figure 4A). Cdx4 overexpression also did not cause any significant changes in the expression levels of the differentiation- and mineralization-related markers, including RUNX2, OSX, OCN, and ALP in the CNCC/Cdx4 cells.

To determine whether elevated levels of *HOX* transcripts affect CNCC differentiation, we subjected CNCC/control and CNCC/Cdx4 cells to in vitro differentiation using mineralization media in the presence of BMP2 for 5 days and 10 days. Real-time PCR analysis determined that the in vitro mineralization induction of cells for 5 days or 10 days led to a significant increase in the transcript levels of the key mineralization regulator genes, RUNX2 and OSX, in both the CNCC/Cdx4 cells and the CNCC/control cells (Figure 4C,D). However, the rate of increase in gene expression over the 10-day period was significantly lower in the CNCC/Cdx4 cells compared to the CNCC/control cells, indicating a decrease in mineralization response in CNCCs upon Cdx4-mediated *HOX* upregulation. Similarly, the transcript levels of the bone marker gene, OCN. were also significantly less upregulated in Cdx4-expressing CNCC compared to the control cells, especially at the 10-day time point (Figure 4E). In contrast to RUNX2, OSX, and OCN, the ALP transcript levels were initially higher at the 5-day time point in CNCC/Cdx4 cells compared to the controls cells and subsequently were comparable in the two cell types with no apparent difference at the 10-day time point (Figure 4F). We also verified the sustained expression of CDX4 in CNCC/Cdx4 cells over the 10-day differentiation period (Figure 4B). 

### 3.5. Elevated HOX Gene Expression and Decreased H3K27me3 Marks on HOX Promoters of Human Skeletal Bone Osteoblasts Compared to Craniofacial Alveolar Bone Progenitors

There is a remarkable difference between the role of *HOX* genes in the patterning of the mesoderm-derived axial and appendicular skeletal bone and the lack of *HOX* gene involvement in the formation of the craniofacial skeleton [50,52,53]. We hypothesized that different “codes” of histone modifications direct the differential expression of *HOX* genes in cranial CNCC-derived AB cells versus mesoderm-derived skeletal osteoblasts. To investigate this possibility, we first compared *HOX* gene expression between AB cells and long bone-derived human osteoblast progenitors (HOb), which served as a comparable mesodermal bone-committed progenitor lineage. Real-time PCR-based gene expression analysis demonstrated that all *HOX* genes analyzed from the *HOXA* and *HOXB* clusters were expressed at significantly higher levels in osteoblasts (HOb) when compared to alveolar bone cells (AB) (Figure 5A). In support of our gene expression data, the ChIP analyses of the HOXA3, HOXA9, HOXB3, and HOXB9 promoter regions were marked with high levels of H3K27me3 repressive histone modification marks in the AB cells when compared to the HOb cells (Figure 5B–E). While our ChIP assays were able to detect the comparatively reduced enrichment for H3K27me3 at the HOXB3 and HOXB9 promoters in HOb cells, H3K27me3 enrichment was completely absent in these cells on the HOXA3 and HOXA9 promoter regions when compared to the AB cells (Figure 5B,C). Interestingly, while H3K4me3 enrichment was significantly higher in the HOb cells for the HOXA3 and HOXA9 promoter regions, the enrichment for this active mark was slightly lower in the HOb cells at the HOXB3 and HOXB9 promoters (Figure 5D,E). These results suggest that the H3K27me3 histone modifications play a primary role in the regulation of *HOX* gene expression in human osteoblast lineages, while H3K4me3 has a fine-tuning effect on select *HOX* promoters. 

### 3.6. Small Molecule-Mediated EZH2 Inhibition Leads to Increased Patterning and Supernumerary Cusp Formation in Mouse Tooth Organs

Our epigenetic analysis and gene expression assays demonstrated that there was an overall higher level of H3K27me3-mediated repression of *HOX* gene promoters in the relatively less differentiated DF cells compared to the further committed AB cells. 

These results suggest that *HOX* gene epigenetic repression is an early event during dental progenitor specification, followed by the subsequent lineage specific activation of select *HOX* genes during differentiation. To investigate whether *HOX* gene silencing by H3K27me3 histone modifications was essential for tooth morphogenesis, *HOX* gene expression was reactivated in ex vivo cultured mouse molar organs with the selective EZH2 inhibitor GSK126. EZH2 catalyzes histone H3K27me3 marks as part of the Polycomb repressive complex 2 (PRC2). GSK126 is a highly selective small molecule inhibitor that decreases H3K27me3 levels globally, leading to a reactivation of PRC2-silenced genes [54]. Mouse tooth germs derived from late cap stage molars were grown in ex vivo cultures for a period of 10 days, resulting in cytodifferentiation and the deposition of the enamel extracellular matrix, as visualized in the control tooth organs (Figure 6A). Compared to the controls, the GSK126-treated mouse tooth germs demonstrated extensive morphological and structural changes (Figure 6B,D,F). Immunohistochemical analysis demonstrated that the expression levels of both HOXA4 and HOXB6 were higher in the GSK126-treated tooth germs compared to the control tooth germs, which is indicative of *HOX* gene reactivation (Figure 6D,F). The HOXA4 and HOXB6 protein levels were the highest in the outer layer of the epithelium surrounding the tooth germ and were elevated upon GSK126 treatment. Interestingly, the enamel layer was notably thicker in the GSK126-treated tooth organs compared to the control tooth organs (Figure 6G), and the thicker enamel area adjacent to the outer layer of the tooth germs demonstrated the highest level of HOXA4/HOXB6 gene reactivation (Figure 6D,F). Furthermore, the gene expression analysis for the key mineralization-related markers revealed that the expression levels of the mineralization gene, OSX, were significantly lower in the GSK126-treated tooth germs (Figure 6H). The transcript levels of RUNX2 and ALP did not change significantly in mouse molars upon GSK126 treatment (Figure 6H). In support of our immunohistochemical analysis, the transcript levels of HOXA4, HOXB6, and HOXA10 were also significantly higher in the GSK126-treated group (Figure 6H). 

## 4. Discussion

The present study was designed to understand the mechanisms underlying the greatly reduced levels of craniofacial *HOX* gene expression from an epigenetic perspective and their effects on development. Our study was based on a series of histone enrichment profiles from pluripotent stem cells and neural crest cells and odontogenic progenitors from the dental follicle, periodontal ligament, alveolar bone, and cementum, as well as bone marrow mesenchymal stem cells and muscle myoblasts. Th embryonic stem cell (ES), neural crest (CNCC), bone marrow, and muscle myoblast enrichment profiles were obtained from the NCBI database, while the odontogenic lineage profiles were from the ChIP-chip studies generated in our laboratory. Using the ChIP-chip data from our laboratory and the available promoter tracks, the histone modifications on individual *HOX* promoters were compared bioinformatically to deduce the H3K4me3 and H3K27me3 histone modification-mediated epigenetic regulation of gene expression during craniofacial development. To ask whether *HOX* gene epigenetic regulation differed between the craniofacial and the trunk neural crest, histone modification patterns in alveolar bone and trunk osteoblasts were compared. To epigenetically modulate *HOX* gene expression and determine the effects of repressing the epigenetic *HOX* gene control machinery, *HOX* gene overexpression was directly induced via Cdx4, while the small molecule against the PRC2 complex member EZH2 was used to block the epigenetic silencing of the *HOX* promoter and indirectly activate *HOX*. Together, these studies provide a comprehensive analysis of the effects of the epigenetic repression of the *HOX* genes during mammalian development and a preliminary assessment of the consequences of aberrant *HOX* gene activation. 

Our study identified H3K27me3 as one of the histone modifications responsible for *HOX* gene repression during craniofacial development. *HOX* gene repression occurred most stringently in ES cells and neural crest cells and remained prominent in all the odontogenic lineages studied here, the dental follicle and dental pulp progenitors, the periodontal ligament fibroblasts, the alveolar bone osteoblasts, and the cementoblasts. The trimethylation of H3 lysine 27 (H327me3) is mainly accomplished by the catalytic portion of the Polycomb repressive complex 2 (PRC2), EZH2, resulting in cell-type specific gene repression and the formation of facultative heterochromatin [55,56]. It is not clear how PRC2 selects its targets, but recent studies have implicated a regional chromatin context, including histone modifications, DNA methylation, chromatin structure, and nuclear organization in the PRC2 target selection [55]. Our ChIP-chip data with their broad repressive marks along all the *HOX* gene clusters have demonstrated that craniofacial *HOX* genes in the neural crest cell populations and odontogenic lineages are typical H3K27me3-rich regions that may act as “super-silencers” [56] that act in concert to achieve a wide-ranging repression of target genes. Our observations are in support of previous studies establishing a crucial role for H3K27me3 histone modification in the silencing of *HOX* genes [9,57,58]. In our study, *HOX* gene cluster silencing through H3K27me3 modifications does not appear to be a rigid block but rather a fine-tuned event designed to permit rudimentary levels of *HOX* gene expression. 

Our data indicate that *HOX* gene promoters were occupied by bivalent histone trimethylation domains in odontogenic progenitors, featuring a dominant H3K27me3 mark that is somewhat balanced by an active H3K4me3 mark. In previous studies, we reported on the bivalent characteristics of odontogenic progenitor promoters undergoing changes in histone trimethylation mark configuration during development and differentiation [35,59,60]. Bivalent histone modifications are considered to set up genes for activation during lineage commitment by H3K4me3 and repress lineage control genes to maintain pluripotency by H3K27me3 [61]. The role of bivalent histone modifications at the promoters to selectively facilitate activation or repression was originally supported by studies co-localizing both H3K27me3 and H3K4me3 at developmentally important promoters [44,60]. A bivalent mark comprising H3K4me2/H3K27me3 enrichment on positional genes, including several *HOX* genes, was identified to play a key role in maintaining the plasticity and positional identity of post-migratory CNCCs [62]. In spite of the presence of both active and repressive histone marks in most progenitors, our studies document an overall higher prevalence of H3K27me3 marks on the *HOX* promoters (Figure 7). *HOX* gene promoter modifications in odontogenic progenitors are therefore reminiscent of bivalent *HOX* domains in ES cells, where the H3K27me3 enrichment values were several times higher than for H3K4me3 [45]. Moreover, our description of H3K27me3/H3K4me3 distribution in odontogenic progenitors as bivalent also stems from other studies which report a dominant role for H3K27me3 in repressing the transcriptional activities of the bivalent genes [63]. Most recently, it has been proposed that bivalent domains and associated chromatin-modifying complexes of the Trithorax and Polycomb group function to safeguard proper and robust differentiation [64], explaining the repression of *HOX* gene expression demonstrated in the present study through bivalent H3 K4/K27 trimethylation. On the other hand, this bivalent mode of *HOX* gene regulation was lost in a differentiated cell type such as keratinocytes (NHEK), with relatively higher levels of H3K4me3 marks over H3K27me3 marks, especially for the *HOX* A and *HOX* C clusters (Figure 2A). Indeed, it was demonstrated that human fetal skin did express *HOX* gene transcripts, including high levels of HOXA4, HOXA5, and HOXA7 [65], all of which were predominantly enriched for H3K4me3 histone marks in our analysis. 

The *HOX* transcript levels in the skeletal osteoblasts were many times higher in the trunk osteoblasts compared to the alveolar bone osteoblasts, and the K27me3 repressive marks on the *HOX* promoters were substantially and significantly elevated in the alveolar bone osteoblasts when compared to the trunk osteoblasts. While microanatomically, biochemically, and biomechanically largely similar, craniofacial and trunk skeletal bones are distinguished by differences in origin and development. Specifically, craniofacial bones develop from the craniofacial neural crest of the neuroectoderm while trunk bones develop from the mesoderm [66]. Moreover, craniofacial bones feature intramembranous mechanisms of ossification in addition to the endochondral mechanism that commonly occurs in trunk skeletal tissues [67]. Cementing differences between the trunk and craniofacial bones, genetic studies have established a predominant role of DLX and MSX genes as pattern regulators in the craniofacial region, while *HOX* genes control pattern formation along the vertebrate body axis from the second pharyngeal arch to the caudal tip. A pivotal study on the role of the PRC2 complex member EZH2 in mouse craniofacial development sheds light on the repression of *HOX* genes in normal craniofacial development as a means to inhibit an osteochondrogenic differentiation program in the craniofacial region [9]. They demonstrate that the Ezh2 conditional knockout prevents the rostral aspect of craniofacial bone and cartilage development by unleashing a dormant *HOX* expression mechanism not active in normal development [9]. The remarkable 5–10-fold higher levels of EZH2 expression in dental pulp cells vs. any of the other odontogenic progenitors (dental follicle, periodontal ligament, alveolar bone, and cementum) suggest high levels epigenic repression involved in various aspects of odontogenesis and dental pulp maintenance. However, similar levels of K27 in the *HOX* cluster of those progenitors suggests that these different levels of EZH2 do not relate to *HOX* regulation.

Previous studies have demonstrated that aberrant craniofacial *HOX* gene expression leads to developmental defects [10,68], and our study suggests that normal craniofacial development requires the presence of repressive H3K27me3 histone marks on craniofacial *HOX* gene promoters. Exploring the use of *HOX* gene de-repressors to explain *HOX* gene function in the craniofacial region, we resorted to Cdx4 overexpression to directly induce *HOX* gene upregulation and the EZH2 inhibitor GSK126 to increase the *HOX* via de-repression. While enhanced levels of *HOX* genes in craniofacial tissues only occur under experimental conditions such as the application of retinoic acid, they have been reported to be associated with major craniofacial birth defects [69,70,71,72,73]. In our studies, *HOX* gene upregulation resulted in decreased mineralization marker gene expression in mouse neural crest cells and in altered patterning during tooth cusp formation in organ culture. So far, both of our Cdx4 overexpression and GSK126 treatment approaches in mouse tooth organs resulted in *HOX* gene upregulation. However, given the varying effects of *Cdx4* overexpression and GSK126 treatment on multiple dental lineage differentiation pathways, additional *HOX*-specific approaches may be used to verify the *HOX*-specific effects on developing tooth germ layers. Changes in mineralization potency and patterning defects both would explain the severe craniofacial phenotype observed in the Ezh2 mutant mice [9] and provide an evolutionary explanation for the striking repression of *HOX* genes in the “new” vertebrate head that features so prominently in vertebrate evolution. Perhaps *HOX* genes evolved in vertebrates through clustering and genome duplication and via collinear Wnt repression to specify regional identity along the vertebrate body axis [14,74,75,76,77]. With the emergence of the vertebrate neural crest [78,79], a new set of gene regulatory networks evolved to coordinate processes such as neural crest specification, migration, and diversification, including transcription factors such as Msx1 and 2, Dlx5 and 6, Pax3 and 7, and FoxD3, as well as Sox5, 6, and 9 [80]. Together, these transcription factors allow for a nuanced fine tuning and coordinated regulation of gene expression that not only coordinate the emigration of neural crest cells but also the complexities involved in the formation of many sophisticated craniofacial organs, including salivary glands, eyes, and teeth. 

## Figures and Tables

**Figure 1 genes-14-00198-f001:**
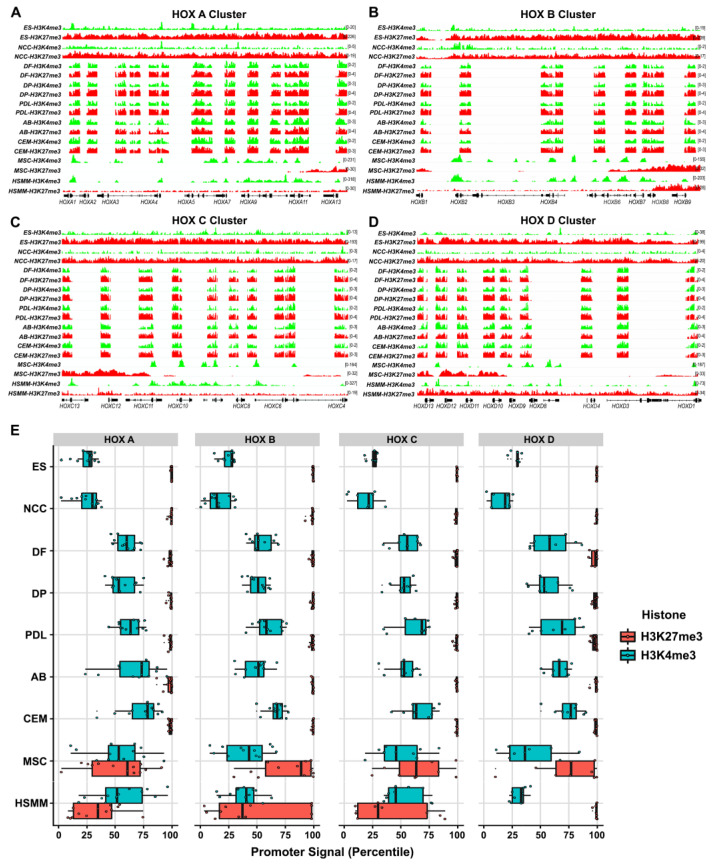
Histone modification profiling at *HOX* promoters in human progenitor cells. (**A**–**D**) IGV genome browser snapshot of H3K4me3 (green track) and H3K27me3 (red track) enrichment for all *HOX* genes from *HOXA* (**A**), *HOXB* (**B**), *HOXC* (**C**), and *HOXD* (**D**) clusters in embryonic stem cells (ES), neural crest cells (CNCC), dental follicle (DF), dental pulp (DP), periodontal ligament (PDL), alveolar bone (AB), cementum (CEM), mesenchymal stem cells (MSC), and human skeletal muscle myoblasts (HSMM). Normalized ChIP-chip replicate data were merged and visualized for each histone modification for DF, DP, PDL, AB, and CEM progenitors. Normalized reads from ChIP-seq histone enrichment data for ES, CNCC, MSC, and HSMM were processed from existing datasets in IGV server and integrated into the combined IGV view. Individual *HOX* gene features are illustrated at the bottom for each *HOX* cluster. (**E**) Box plot depicting H3K4me3 and H3K27me3 promoter signals at the cluster level for each progenitor type across all *HOX* clusters. Individual *HOX* genes are represented as colored dots within each cluster. The box size indicates the distribution of histone enrichment data among *HOX* promoters from each cluster for histone modifications as indicated, and the median for each dataset is represented as a vertical line within each box. ES, CNCC, DF, DP, PDL, AB, and CEM progenitors exhibit the highest levels of H3K27me3 repressive marks across all *HOX* clusters.

**Figure 2 genes-14-00198-f002:**
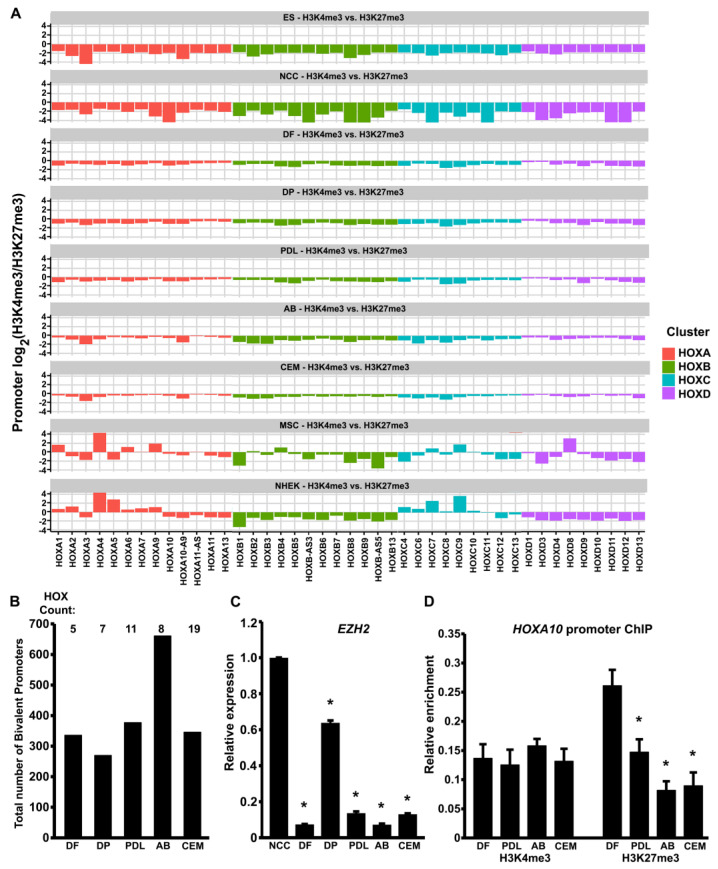
*HOX* gene promoters are marked by a bivalent signature in odontogenic progenitors. (**A**) Bar plot comparison of log_2_ ratio of H3K4me3/H3K27me3 enrichment values for each *HOX* promoter in progenitors as indicated. Bars are color-coded to demarcate each *HOX* cluster. Negative values are indicative of H3K27me3 predominance at the promoter, while positive values imply higher H3K4me3 enrichment. *HOX* gene promoters were generally characterized by higher levels of H3K27me3 marks and a much lower level of H3K4me3. ES cells and odontogenic progenitors exhibited a promoter enrichment profile similar to a bivalent signature in our analysis. (**B**) Comparison of the total number of bivalent promoters in odontogenic progenitors obtained from ChIP-chip analysis. The number of *HOX* genes featured among the bivalent promoters is indicated at the top. (**C**) Comparison of EZH2 transcript levels in CNCCs and odontogenic progenitors. Transcript levels in all odontogenic progenitors were compared individually against CNCCs. Data were compiled from biological replicates, n = 6. (**D**) Representative ChIP analysis for H3K4me3 and H3K27me3 enrichment at the *HOXA10* promoter in odontogenic progenitors as indicated. Enrichment from no antibody/beads immunoprecipitation was subtracted from corresponding antibody enrichment values before graphing. Enrichment levels in PDL, AB, and CEM were compared to DF cells. Levels of H3K4me3 enrichment were almost identical in all progenitors, while H3K27me3 promoter enrichment was the highest in DF and lowest in AB progenitors. Data were compiled from five independent biological replicates. Differences were considered as statistically significant (*) at *p* < 0.05 for (**C**,**D**).

**Figure 3 genes-14-00198-f003:**
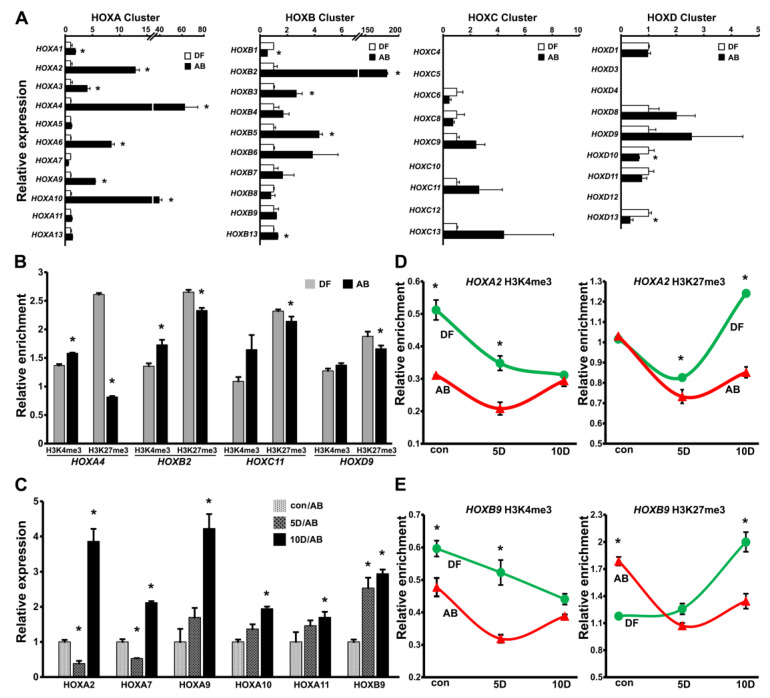
Higher *HOX* gene expression in AB cells when compared to DF progenitors. (**A**) Quantitative PCR analyses of *HOX* gene expression in human DF and AB progenitors. Transcript levels in AB cells were compared against DF cells and graphed as fold difference in expression levels. Expression data were obtained from biological replicates, n = 7. (**B**) Representative ChIP analysis for H3K4me3 and H3K27me3 enrichment in DF and AB progenitors at *HOXA4*, *HOXB2*, *HOXC11*, and *HOXD9* promoter regions. Histone modification enrichment values were normalized against corresponding inputs for each promoter primer pair after subtracting the background enrichment values. Enrichment data in AB cells were compared against DF cells. Data are from biological replicates, n = 5. (**C**) Comparison of select *HOX* gene expression levels as determined by quantitative PCR in control AB cells (con/AB) and in AB cells after 5 days (5D/AB) or 10 days (10D/AB) of in vitro mineralization induction. Expression levels at the 5 day and 10 day time point were compared individually to the con time point, n = 6. (**D**,**E**) Visual comparison of H3K4me3 and H3K27me3 enrichment dynamics at *HOXA2* and *HOXB9* promoters between DF and AB progenitors after 5 (5D) and 10 days (10D) of in vitro mineralization induction. Non-induced cells (con) were used to determine baseline levels of histone modification enrichment. Background enrichment values were subtracted in each case from corresponding histone enrichment values, n = 5. Elevated *HOX* gene expression was accompanied by a reduction in H3K27me3 enrichment over the 10-day induction period in AB cells. All assays were performed with early passage DF and AB progenitors (passage 3–4). Expression and ChIP data are presented as the mean ± standard deviation of the mean and considered as statistically significant (*) at *p* < 0.05.

**Figure 4 genes-14-00198-f004:**
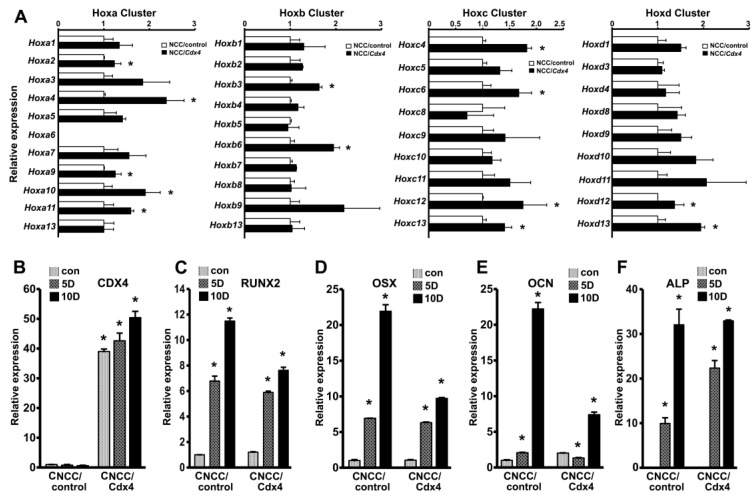
Cdx4-mediated *HOX* gene activation suppresses CNCC mineralization potential. (**A**) q-PCR comparison of *HOX* gene expression in mouse CNCC control cells (CNCC/control) and CNCC stably expressing CDX4 (CNCC/CDX4). Expression data are from biological replicates, n = 7. (**B**) Comparison of CDX4 transcript levels in CNCC/control and CNCC/Cdx4 cells during the course of in vitro mineralization induction. CDX4 transcript levels were consistently high in the CNCC/Cdx4 group throughout the experiment. Expression levels at each time point were compared to CNCC/control uninduced cells. (**C**–**F**) Comparison of mineralization-associated gene expression, including (**C**) RUNX2, (**D**) OSX, (**E**) OCN, and (**F**) ALP in CNCC/control and CNCC/Cdx4 cells after 5 or 10 days of in vitro mineralization induction. Expression at the 5-day and 10-day time point were separately compared to the corresponding control group and statistical significance determined, n = 7. Non-induced cells (con) from each group were used as controls. Differences were considered as statistically significant (*) at *p* < 0.05.

**Figure 5 genes-14-00198-f005:**
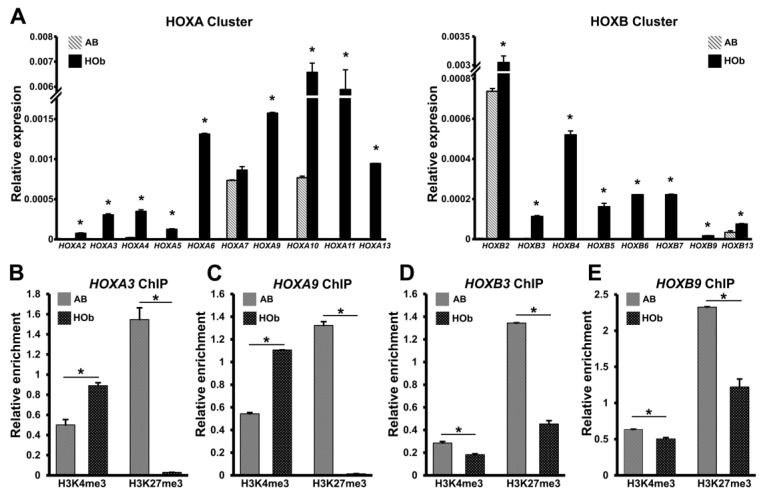
Comparison of *HOX* gene expression in craniofacial bone and skeletal bone progenitors. (**A**) q-PCR analysis comparing *HOX* gene expression from *HOXA* and *HOXB* clusters in human alveolar bone progenitors (AB) and human osteoblasts (HOb). Absolute values were graphed to accommodate the vast differences in gene expression levels. Expression data are from biological replicates, n = 7. *HOXA* and HOXB gene transcript levels were significantly elevated in osteoblast progenitors. (**B**–**E**) ChIP analysis to quantify H3K4me3 and H3K27me3 enrichment in AB and HOb cells at *HOXA3* (**B**), *HOXA9* (**C**), *HOXB3* (**D**), and *HOXB9* (**E**) promoter regions. Histone modification enrichment was normalized against enrichment from input chromatin for each *HOX* primer pair after subtracting the corresponding background enrichment values, n = 5. H3K27me3 enrichment was significantly lower for all *HOX* promoters analyzed in HOb cells compared to AB cells. Differences were considered as statistically significant (*) at *p* < 0.05.

**Figure 6 genes-14-00198-f006:**
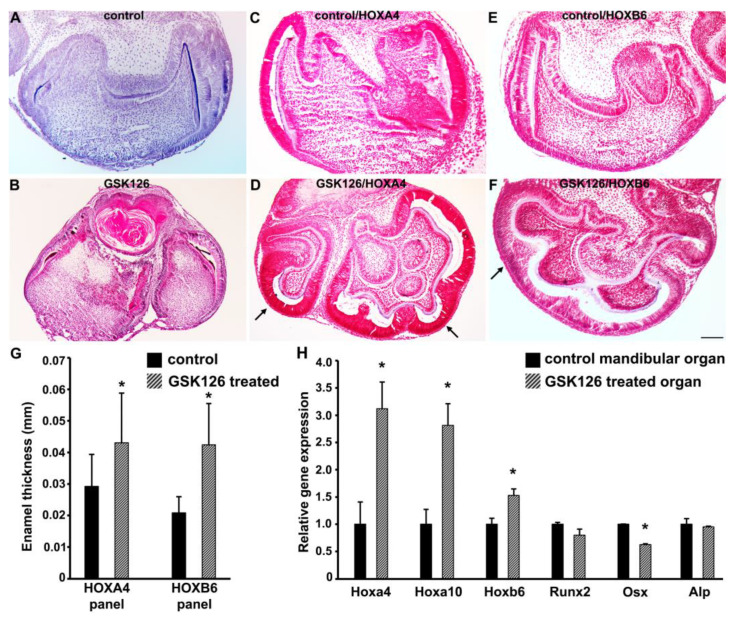
Small molecule-mediated *HOX* gene reactivation in mouse tooth organs. (**A**–**F**) Immunohistochemical analysis of control and GSK126-treated mouse tooth germs. (**A**) Hematoxylin-stained control mouse tooth organ serving as a negative control for *HOX* antibody detection. (**B**) Representative mouse tooth organ treated with GSK126 for 10 days, stained using Masons Trichrome stain. (**C**–**F**) Immunohistochemical detection of HOXA4 and HOXB6 expression in control tooth organs (**C**,**E**) and GSK126-treated tooth organs (**D**,**F**). Sites of distinct immunohistochemical reactions are marked with an arrow in (**D**,**F**). Sections were counterstained with hematoxylin and visualized. (**G**) Comparison of enamel thickness between control and GSK126-treated tooth organs from panels (**C**–**F**). Fifty to one hundred measurements were performed on each sample for each experimental condition using Image J. **(H**) qPCR analysis comparing expression levels of *HOX* genes and key mineralization-related genes between control and GSK126-treated tooth organs. Samples for gene expression comparison were obtained from more than 6 individually treated tooth germs. Differences were considered as statistically significant (*) at *p* < 0.05. Scale, 0.1 mm.

**Figure 7 genes-14-00198-f007:**
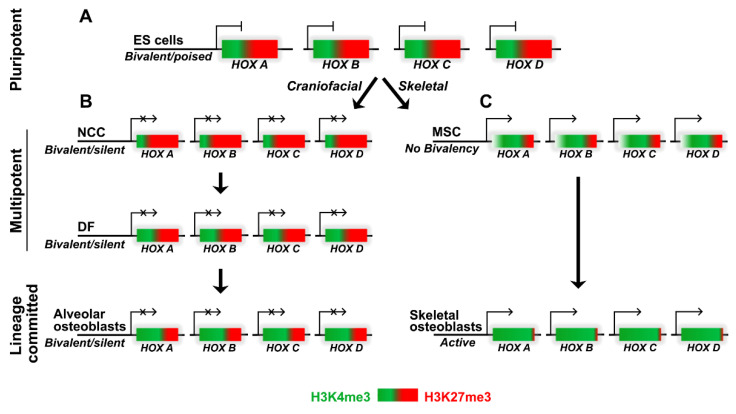
*HOX* gene transcriptional regulation by histone modifications. (**A**) *HOX* promoters (*HOX* A, *HOX* B, *HOX* C, and *HOX* D) in embryonic stem (ES) cells are transcriptionally poised and feature bivalent chromatin comprising H3K4me3 (green) and H3K27me3 (red) histone modifications. (**B**) Craniofacial lineage progenitors, neural crest cells (NCC), dental follicle (DF), and alveolar osteoblasts do not express *HOX* genes and exhibit varying levels of bivalency. (**C**) *HOX* promoter bivalency is lost in the skeletal lineage mesenchymal stem cells (MSCs) and in lineage-committed skeletal osteoblasts, leading to high levels of *HOX* gene expression.

## Data Availability

Data are held at the TAMU Center for Craniofacial Research and Diagnosis and are available upon request.

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
