# Peer review of "Changes in Hox Gene Chromatin Organization during Odontogenic Lineage Specification"

_genes, 2023, doi:10.3390/genes14010198_

Round 1

Reviewer 1 Report

Manuscript Genes- 2110007, review

 Changes in Hox gene chromatin organization during odontogenic lineage 2 specificatio

By Gopinathan et al.

The authors used two chromatin modifiers to analyze HOX gene expression in different cells derived from the CNCCs. They correlated the present of these modifiers on HOX gene expression related to the differentiation state of the different cells they analyzed. ChIP-ChIP experiments were used to understand the presence of these modifiers on proximal promoters. This research is well within the expertise of the researchers and an extension of their previous work. The manuscript is well-written and logical with good data. 

Comments:

line 148-149-- Not clear what was sonicated, can nuclei be sonicated to 300bp?? I think the authors meant chromatin is sonicated to 300bp-1kb fragments.

line 658-- "Our study identified H3K27me3 as the key histone modification"-- This is not correct as other modifiers were not tested, thus they cannot make this claim. Please revise this statement 

Author Response

We would like to sincerely thank this reviewer for his/her time and for expert suggestions to revise our manuscript.  Please find our response to individual concerns below. 

  1. Line 148-149: It is not clear whether nuclei or chromatin is being sheared for ChIP reactions

We thank the reviewer for allowing us to clarify this point.  The sentence has now been modified to “Nuclei from formaldehyde treated cells were lysed in cold lysis buffer and chromatin sonicated to a size of 300bp-1kb in a cup horn sonicator”.

  1. Line 658: Statement identifying H3K27me3 as the key histone modification responsible for Hox repression needs to be revised

We appreciate the reviewer’s concern and agree that data from other histone modifications need to be taken into consideration to truly understand the complexity of HOX promoter epigenetic regulation.  We have modified the statement to indicate that H3K27me3 is one of the histone modifications responsible for HOX gene repression during craniofacial development.  We have also added that other histone modifications will be tested in future studies. 

Reviewer 2 Report

In this study, the authors investigated the epigenetic regulation of Hox genes from undifferentiated neural crest cells to semi-differentiated odontogenic progenitors using the ChIP-on-chip approach to determine why expression of Hox genes is down-regulated during neural crest cell differentiation. They demonstrated high levels of repressive H3K27me3 (K27) marks on Hox gene promoters in ES and cranial neural crest cells comparing to H3K4me3 (K4) marks, while the K4/K27 ratio was less repressive in odontogenic progenitors, dental follicle, dental pulp, periodontal ligament fibroblasts, alveolar bone osteoblasts, and cementoblasts. To determine the effect of elevated expression of Hox genes in craniofacial neural crest cells, they transfected Cdx4 into cells and found a significant decrease in mineralization markers, including Runx2, Osx and Ocn upon Hox elevation. Taken together, these studies demonstrate the significant effects of epigenetic regulation at all stages of the differentiation of craniofacial peripheral tissues from the neural crest, including lineage specification, tissue differentiation, and patterning. 

Overall speaking, this study is very comprehensive and highly informative. The experiments were well designed and conducted. The findings of this study support the authorss conclusion. This study is very helpful to the entire field of craniofacial developmental biology.

Minor comment:

The authors need to disclose what type of cells was used in Cdx4 transfection experiment. 

Author Response

We would like to sincerely thank this reviewer for his/her time and for generous comments in support of our manuscript.

Minor comment: Authors need to disclose the cell type used for Cdx4 transfections

We used the O9-1 mouse cranial neural crest cells (CNCC) for the Cdx4 transfection experiment. The cell type and the source are now mentioned in the methods section.  We have now added this information in the methods sub-section corresponding to Cdx4 plasmid constructs and transfections.

Reviewer 3 Report

In this study, authors tried to define the epigenetic events that regulates Hox gene expression from undifferentiated neural crest cells to semi-differentiated odontogenic progenitors. In addition, they also examined effects of Hox gene overexpression on the neural crest lineage differentiation by Cdx4 transfection and Ezh2 inhibitor.

Indeed, epigenetic regulation of Hox genes in neural crest cells has been already extensively studied, and authors’ data are largely just confirming previous knowledge. However, they provided specific implication in the context of dental lineage, which may be interesting for specialists.

Specific comments:

1.       From line 51, ‘A direct link between HOX gene repression in CNCC and bone and cartilage formation in the craniofacial skeleton was also demonstrated in mice subjected to conditional inactivation of the Polycomb repressive complex subunit, Ezh2 [9].’ This study did not provide a ‘direct’ link because they did not carry out HOX gene inactivation in Polycomb knock-out mice. They proved just a correlation, and thus authors should tone down their statement (also related to my comment 9).

2.       From line 324, ‘Among the dental lineages, DF, DP and PDL cells exhibited the highest levels of HOX promoter enrichment for H3K27me3, while AB and CEM cells demonstrated a comparatively lower level of H3K27me3 enrichment across all HOX clusters. Notably, when compared to ES and CNCC cells, the active H3K4me3 marks on HOX promoters were higher in all odontogenic progenitor lineages, with AB and CEM cells exhibiting the highest levels.’ This is an over-simplification of interpretation regarding ChIP- chip data. Even though there are some HOX genes (e.g., HOXA4) that show such dynamics, I do not necessarily see the overall tendency described by authors. Especially I do not see any difference regarding H3K27me3 levels among ES, NC, DF, DP, PDL, AB and CEM in Fig. 1E.

3.       Authors must provide scales for genome browser views in Fig. S1.

4.       From line 343, ‘In contrast, H3K27me3 enrichment signals were highly variable and significantly lower for all HOX clusters in MSCs and HSMMs’, also from line 388, ‘In contrast, the H3K4me3/H3K27me3 ratios in CNCC were significantly lower and trended toward a higher negative value compared to both ES cells and odontogenic progenitors.’ Authors did not show statistical tests to prove significant difference. 

5.       In Fig. 2C, they showed high transcripts of Ezh2 in DP, but did not provide biological interpretation of this.

6.       Lines 451 and 540 ‘Real time expression analysis’ is unclear. Maybe real time PCR analysis of transcripts?

7.       Line 483, data should be shown.

8.       Regarding line 626, ‘Interestingly, the enamel layer was notably thicker in GSK126 treated tooth organs compared to control tooth organs’, quantitative analysis should be carried out.

9.       Overall, authors must be careful regarding the effects of Cdx4 and Gsk126 treatment on differentiation. These treatments can affect expression of numerous genes other than HOX genes, and there are no guarantees that effects observed in the present study are due to overexpression of HOX genes. This point should be clearly discussed and authors should tone down their current statements.  

Author Response

Reviewer 3

We thank this reviewer for his/her time and the comprehensive review of our manuscript.  We sincerely appreciate the encouraging comments and suggestions.  Here we are addressing each of the specific comments raised by the reviewer below.

  1. Statement mentioning “A direct link between HOX gene repression in CNCC and bone and cartilage formation in the craniofacial skeleton” needs to be corrected

Thank you for pointing out this inaccuracy.  We have now replaced “direct link” with “correlation”.  This wording more accurately reflects the relationship between HOX gene expression and craniofacial skeletogenesis in Ezh2 conditional mice as described in the Schwarz et al. paper.

  1. Concerns regarding an overly simplistic interpretation of ChIP-chip data when comparing odontogenic progenitors and other cell lineages

We agree with this reviewer’s observation.  The primary goal of our comprehensive histone modification data comparisons using IGV was to identify overarching trends in enrichment levels of H3K4me3 and H3K27me3 histone modifications among progenitors from different lineages.  Our analysis indicated that HOX promoter regulation as it relates to H3K4 and H3K27 histone methylation is predominantly regulated at the cluster level.  However, we do appreciate the concerns raised by the reviewer and agree that a unilateral interpretation of such complex data cannot account for the differences in histone modification dynamics at individual HOX promoters within each cluster.  Therefore, we have re-written this part of the Results section (Figure 1 A-D) with more specific details to clearly identify HOX promoters that show a difference among the various progenitors.  We have now removed the statement in Fig. 1E.

  1. Scales for IGV plots in Fig. S1

We have rectified this error and are uploading a new version of this figure with histone enrichment scales.

  1. Statements in line 343 and 388 mention significant changes in histone enrichment but lack statistical tests

The histone enrichment comparisons in Figure 1E and H3K4me3/H3K27me3 enrichment ratios in Figure 2A are primarily designed to identify overall trends in histone enrichment among different lineages.  Similar to other IGV based comparisons within Figure 1 A-E and in Figure 2A, these were not subjected to statistical significance tests.  We thank the reviewer for pointing out this error to us and have corrected both sentences in the corresponding results sections to remove any confusion.

  1. Biological interpretation for high transcript levels of Ezh2 in Dental Pulp progenitors is missing

We agree with this reviewer on this unexpected finding.  We have now added a statement suggesting that “The remarkable 5-10 fold higher levels of EZH2 expression in dental pulp cells vs. any of the other odontogenic progenitors (dental follicle, periodontal ligament, alveolar bone, cementum) suggest high levels epigenic repression involved in various aspects of odontogenesis and dental pulp maintenance.  However, similar levels of K27 in the Hox cluster of those progenitors suggests that these different levels of EZH2 do not relate to Hox regulation.” 

  1. Lines 451 and 540: “Real time expression analysis” is unclear

We have modified the respective sentences to – “Real time PCR analysis”.

  1. Data should be shown for Line 483

We have uploaded a separate figure (Supplemental Figure S2) quantifying gene expression change for select HOX genes in DF cells after 5 and 10 days of mineralization induction. 

  1. Quantitative analysis of enamel thickness should be carried out

Based on the reviewer’s suggestion we have performed quantitative measurements of enamel layer thickness in both control and GSK126 treated mouse tooth organs using Image J software. We are now presenting this data as an addition to Figure 6 (Figure 6G). The corresponding figure legend and results were also updated to match the new figure. 

  1. Potential side effects resulting from Cdx4 overexpression and GSK126 treatment should be discussed

We appreciate the concern raised by the reviewer and are aware of the complexities involved in interpreting data from our Cdx4 expression and GSK126 inhibitor studies.  We have modified the discussion to clearly discuss this aspect. 

Round 2

Reviewer 3 Report

Authors addressed all the comments.